# Tablet Technology for Writing and Drawing during Functional Magnetic Resonance Imaging: A Review

**DOI:** 10.3390/s21020401

**Published:** 2021-01-08

**Authors:** Zhongmin Lin, Fred Tam, Nathan W. Churchill, Tom A. Schweizer, Simon J. Graham

**Affiliations:** 1Department of Medical Biophysics, Faculty of Medicine, University of Toronto, Toronto, ON M5G 1L7, Canada; zhongmin.lin@mail.utoronto.ca; 2Physical Sciences, Sunnybrook Research Institute, Toronto, ON M4N 3M5, Canada; fred.tam@sri.utoronto.ca; 3Keenan Research Centre for Biomedical Science, St. Michael’s Hospital, Toronto, ON M5B 1T8, Canada; nchurchill.research@gmail.com (N.W.C.); Tom.Schweizer@unityhealth.to (T.A.S.); 4Division of Neurosurgery, St. Michael’s Hospital, Toronto, ON M5B 1W8, Canada

**Keywords:** fMRI, fMRI compatibility, fMRI equipment, computerized tablet, neuropsychological tests, functional neurosurgery, essential tremor, writing, drawing, kinematics

## Abstract

Functional magnetic resonance imaging (fMRI) is a powerful modality to study brain activity. To approximate naturalistic writing and drawing behaviours inside the scanner, many fMRI-compatible tablet technologies have been developed. The digitizing feature of the tablets also allows examination of behavioural kinematics with greater detail than using paper. With enhanced ecological validity, tablet devices have advanced the fields of neuropsychological tests, neurosurgery, and neurolinguistics. Specifically, tablet devices have been used to adopt many traditional paper-based writing and drawing neuropsychological tests for fMRI. In functional neurosurgery, tablet technologies have enabled intra-operative brain mapping during awake craniotomy in brain tumour patients, as well as quantitative tremor assessment for treatment outcome monitoring. Tablet devices also play an important role in identifying the neural correlates of writing in the healthy and diseased brain. The fMRI-compatible tablets provide an excellent platform to support naturalistic motor responses and examine detailed behavioural kinematics.

## 1. Introduction

Since the seminal papers were published describing functional magnetic resonance imaging (fMRI) of brain activity in the human visual cortex and motor cortex in the early 1990s, first using exogenous contrast agents and then using the blood oxygenation level-dependent (BOLD) mechanism [1,2,3], fMRI has revolutionized neuroscience research and the understanding of brain health and disease. The fMRI method has seen many advances in the past three decades, including specialized sequences with enhanced spatiotemporal resolution, and novel image processing algorithms to obtain maps of brain activity with enhanced signal detection power. At the same time, fMRI research has progressed to probe increasingly complex aspects of human behaviour using task-based imaging. The present review article describes some of the technical development work and applications that have arisen from this latter pathway of research, focusing on behaviour that involves writing and drawing movements.

The vast majority of task-based fMRI studies investigate brain activity associated with the presentation of relatively simple visual or auditory stimuli, using simple button presses to indicate decisions and responses. This is partly due to the long history in experimental psychology of administering such tasks using standard computer peripherals (the display screen, headphones and keyboard). These devices can be used creatively to build up a picture of how the brain operates in the complex, daily world as demonstrated through the many reductionist fMRI experiments that currently exist. Historically, cognitive psychology has focused on determining the key basic units of cognitive processes, such as the classic working memory model of Baddeley, for example, that involves central executive control, an articulatory loop and a visuospatial scratchpad [4]. Reductionist task-based fMRI experiments align with testing such models, by contrasting simple tasks with subtle differences meant to isolate the neural substrate of the model components. By combining the knowledge gained from many such experiments, a more complete understanding of human brain function can be achieved. However, there is also a need to undertake more complicated fMRI experiments. For example, brain activity associated with certain forms of human communication (e.g., speaking and writing) is not ideally represented by experiments involving button press responses. In addition, more naturalistic fMRI experiments are required to understand the extent that the reductionist viewpoint is valid, and also to reveal how various mental processes ebb and flow dynamically during the performance of complex tasks.

In this context, the primary focus of this review is on the optimization of fMRI-compatible devices and approaches to study the brain activity associated with writing and drawing, and real-world tasks that involve this form of interaction. (Devices and approaches to study overt speech are also interesting, but fall outside the scope of this narrative [5,6].) Various device considerations are reviewed first, culminating in a description of computerized “tablet system technology”, which, through use of a touch-sensitive surface and accompanying stylus, permits effective fMRI studies of writing and drawing behaviour. This is followed by a review of various fMRI applications where tablet systems have been utilized, and a discussion of how this technology and research are likely to progress in the future.

## 2. Tablet Technology for fMRI Studies of Writing and Drawing

Deferring an extended discussion of the scientific and clinical imperatives for studying the brain activity associated with writing and drawing, a useful starting point for discussing how tablet technology has developed is simply to consider the practicalities of enabling (and also recording) such behaviour during an fMRI experiment. The fMRI environment imposes multiple constraints. The imaging participant must lie supine within the narrow bore of the magnet (60–70 cm in diameter), and must remain sufficiently still such that movement of their head does not confound the data collection, usually over multiple collection periods (each of about 10 min) to obtain reliable measurements of brain activity. Any ancillary device that is present must experience negligible magnetic forces, negligible radiofrequency heating, and introduce negligible levels of radiofrequency interference such that the device operates properly with no impact on the temporal signal-to-noise ratio (SNR) of the fMRI data. Performing task-based fMRI of real-world writing and drawing behaviour is thus extremely difficult in this context. Even if a configuration could be found that allowed the participant to view what they were writing or drawing with pen and paper, this would become cramped and uncomfortable over time. Furthermore, it is not immediately clear how to cue the onset and offset of repeated task performance precisely [7], how to record the performance quantitatively, and how to replenish the paper supply intermittently. Given these challenges, it is not surprising that fMRI researchers have attempted to bypass them by using other response strategies, such as instructing participants to write and draw mid-air using their index finger [8,9]; adapting the behavioural task in question from a visuo-motor to an auditory-motor construct, such that auditory cues elicit certain verbal articulations [10]; or to resort to task modifications that permit button press responses [11,12].

Two major inter-related problems arise when attempting to interpret the brain activity generated by such bypassing tasks. First, it is usually hypothesized that the brain activity associated with the “core elements” of the task in question are somehow preserved, even though the response mechanism is not as realistic as desired. However, it is not always clear what the core elements are. Second, the lack of a realistic response may turn a simple task, normally performed efficiently by the brain after extensive prior practice in the real world, into a novel task that has higher cognitive and motor demands. In addition to creating uncertainty over whether brain regions responsible for processing the core elements of the task are similarly engaged as they would be in the real-world task, this “response mapping” problem can preferentially engage brain areas involved in cognitive control (e.g., the frontal lobes) rather than those involved in highly skilled coordinated movement (e.g., the cerebellum). Learning or habituation effects may occur, adding temporal effects in the fMRI data. It is not always straightforward to disentangle all of these experimental confounds. Consequently, although the bypass approach discussed above can provide important initial information about brain activity, additional corroborating evidence is required using methods that are much more ecologically valid (i.e., produce behaviour that closely approximates naturalistic performance in the real world).

This conclusion has led researchers to directly confront the challenge of developing a device or approach that approximates naturalistic writing and drawing during fMRI. An additional secondary objective has involved developing methods that enable writing and drawing behaviours to be quantified objectively for detailed analysis of brain–behaviour relationships. Early efforts involved a resistance-based pen movement tracer, although this method was limited by calibration issues and the need to perform a tracing (rather than freeform drawing) movement along a predefined, one dimensional path [7,13]. Another early approach involved a fibre-optic system to track the position of a stylus tip on a two-dimensional surface, although some calibration and position-tracking limitations were identified [7,14].

Building on this work and attempting to overcome the limitations of previous devices, a touch-sensitive digitizing tablet was subsequently developed by members of our group to enable more realistic writing and drawing behaviour during fMRI, including detailed behavioural recording [7,15]. Like computer mice, touch-sensitive tablets are computer input devices that record x,y coordinate values representing the position of contact on a touch-sensitive screen. After initial signal processing, the written/drawn trajectory can be reproduced on the computer display. Consumer electronics versions of computer tablets are now ubiquitous, with the touch screen and display integrated together.

Our fMRI-compatible tablet system does not integrate the touch screen and display, although others have recently investigated this approach in the confined fMRI environment [16]. Instead, the touch-sensitive surface and stylus are mounted in isolation on an elevated support platform that can be attached to the patient table, together with a controller box and the necessary connecting cables, drivers, and software to record responses. This tablet system is meant to be used in conjunction with an fMRI-compatible display system that delivers visual stimuli to a display screen or goggles. Many options for visual stimulus presentation are commercially available, with one configuration consisting of an fMRI-compatible projector mounted at the back of the magnet bore to display task-relevant cues and visual stimuli on a rear-projection screen. The participant views the screen through an angled mirror while lying supine and, in this manner, can perceive tablet interactions overlaid on the display.

The initial tablet prototype (prototype 1, Figure 1) employed a polyester laminate (PL) resistive four-wire transparent touchscreen (Microtouch, Model #RES-6.4-PL4, 3M, Inc., St. Paul, MN, USA; 16 cm diagonal; 13 × 10 cm active area) along with its matching controller board (Microtouch, Model #SC400, 3M, Inc., St. Paul, MN, USA). In addition to its excellent accessibility and affordability (touchscreen and controller cost less than USD 100), as well as straightforward assembly, this hardware was selected for a number of technical reasons. First, by using indium tin oxide resistive coatings and a glass substrate, the PL coversheet is not ferromagnetic and connects very easily to magnetically shielded and filtered cables, providing fMRI-compatibility. Second, the touch-sensitive accuracy (0.005 inches) and report rate (180 reports/s) provide very accurate and detailed capture of writing and drawing behaviour. Third, touch and movement registration are not limited to an fMRI-compatible tool such as a stylus, and can be performed by any reasonable appendage that can be moved into contact with the touch-sensitive surface, such as a finger, knee or foot. In such circumstances, tablet contact can potentially be optimized using wearable fMRI-compatible apparel with a protruding point of contact.

To protect against surface damage and unintentional touches, the touchscreen is mounted into a plastic holding frame that attaches to an optional plastic tablet support platform. The platform can be tilted (up to 90 degrees) and adjusted in height (20 to 40 cm above the patient table) for comfort within the magnet bore, while minimizing interference from the torso and from respiratory motion. A junction box under the tablet frame provides connection to the optional stylus (2-pin) and a shielded cable (Type 9539, Belden Inc., St. Louis, MO, USA) running to the penetration panel in the MRI system. The optional stylus consists of a modified plastic pen barrel (approximately 12 cm in length) with a microswitch on the tip to detect contact between the stylus and touchscreen surface. Pushing with moderate force on the touchscreen with the stylus activates the microswitch, thus producing a small amount of tactile feedback and registering a button press. This optional button input is suitable for response recording or as a crude pressure indicator during task performance. The recorded tablet and stylus signals pass through an EMI (electromagnetic interference) filter (56-705-005-LI, Spectrum Control, Inc., Fairview, PA, USA) at the penetration panel and transmit to the tablet controller box outside the MRI suite via shielded cables. The controller box contains the electronic logic in the tablet controller board, power conditioner, and receptacles for USB (universal serial bus) connections to the fMRI stimulus/response computer. The controller box connects to a computer using USB connections for data transmission from the tablet and stylus to the computer. In addition, during fMRI experiments, the MRI system emits pulse triggers in synchrony with the imaging sequence. These triggers are then converted to keyboard events via a trigger conversion box to initiate the behavioural task onset, thus time-locking the behavioural tracking data with the fMRI acquisition.

As indicated in the research applications below, tablet prototype 1 has been very useful over the years. It has undergone many simple optimizations and revisions, as part of disseminating the system and technology to the scientific community. For example, the stylus microswitch was replaced with a force sensing resistor, and numerous minor mechanical and electronics changes were made for manufacturability, robustness, and to fit in different MRI systems.

Tablet prototype 1 has one feature that may have a significant impact on behavioural performance under some circumstances: the participant is only able to view their tablet interactions, and not their hand manipulating the stylus. Therefore, more reliance is placed on proprioceptive sensation (i.e., sensory receptors in tendons of the hand and arm that help the participant judge where they are making writing and drawing movements on the tablet). This limits the ecological validity and potentially makes it difficult to perform tablet responses that require fine motor skills, or makes it difficult for patients with certain brain dysfunctions (e.g., stroke) to use the tablet. A second-generation prototype (prototype 2, Figure 2) was developed to address this problem by providing visual feedback of hand position (VFHP). Prototype 2 includes a video recording and processing platform, and an augmented reality environment that enables the participant to view the display of the tablet computer overlaid with live video of hand/stylus/touch-surface interactions. Using this approach, participants have increased awareness of their hand position in real time and thus can perform tablet interactions with enhanced ecological validity.

As shown in Figure 2, the additional hardware in prototype 2 consists of an MRI-compatible “TabletCam” colour video camera (12M-i with 4.3 mm lens, MRC Systems, GmbH, Heidelberg, DEU) and its mounting frame; light emitting diodes (LEDs) to illuminate the tablet field of view; and an additional computer outside the magnet room, for video processing with two video capture cards. One capture card is for the TabletCam video feed, and the second card, with an appropriate video format converter, is for the stimulus/response computer feedback. Whereas the original stimulus/response computer is still utilized to present task-related stimuli with precise timing, and to log tablet responses (x, y coordinates and force data as a function of time), drivers and software on the video processing computer are used to fuse segmented video of hand/stylus interactions on the touch-sensitive surface of the tablet, task-related visual stimuli, and graphical representation of the interactions as ink-marks, thus creating an interactive augmented reality environment for subsequent display to the user. The segmented video can be created by various mechanisms, although the simplest is a “blue-screen” approach analogous to what is used when an announcer stands in front of weather maps on television. The tablet surface can be covered in blue tape, which enables a video “mask” to be created of the hand and stylus only (zero signal intensity elsewhere) by segmenting the acquired video based on colour content.

An fMRI study involving nine healthy young adults was conducted to examine the effect of tablet prototypes (with or without VFHP) on writing quality and brain activation [17]. Using simple writing tasks like copying grocery items, phone numbers, and a paragraph, writing with VFHP was found to be less cramped, more clear, and more properly positioned than that without VFHP, as shown in Figure 3 [17].

As summarized in Table 1, the two tablet prototypes offer advantages for studying writing and drawing behaviour during fMRI of brain activity, in comparison to other alternatives (chosen here for example as finger drawing mid-air [8]). Whereas mid-air finger drawing enables realistic motor behaviour and is simple to execute because no fMRI-compatible devices are required, this method does not permit the user to interact with a writing surface, or to receive written performance feedback. Moreover, the approach does not provide behavioural recording. In comparison, both tablet prototypes provide much improved ecological validity combined with detailed digitized behavioural recording, although device hardware is required. The reliance on proprioception, characteristic of performing with tablet prototype 1, is alleviated by the use of VFHP in tablet prototype 2. This improvement inevitably means that some visual stimuli will be obstructed by the virtual hand in the augmented reality environment, as in the real-world—and this effect has to be considered when evaluating fMRI task performance.

Numerous fMRI application studies have been published involving either prototype of this tablet system. Other labs have also developed related technology. For example, a resistive touchscreen for measuring hand kinematics has been developed by Braadbart et al. [18]. Compared to tablet prototype 1 [7], the Braadbart tablet uses the same indium tin oxide as the resistive coating, but it is connected to the controlling computer by optical fibres to minimize electromagnetic interference, rather than by shielded cables [18]. In a study of 18 young healthy adults, the fMRI brain activations evoked by a dynamic drawing task using the Braadbart tablet were shown to be consistent with previous research in writing and drawing kinematics [18,19,20,21]. More recently, the “MRItab”, an fMRI-compatible tablet with integrated touch screen and video display screen for direct VFHP, has been developed and validated for three young healthy adults [16]. The touchscreen overlay of the MRItab is constructed similarly to tablet prototype 1 for its excellent spatial and temporal resolution [7,16]. However, placed perpendicular to the scanner bed, the MRItab requires the user to look out at an extreme angle from the interior of the magnet bore for visual feedback, which might lead to uncomfortable neck posture and, thus, diminished ecological validity.

A number of other tablet devices have been employed in the fMRI literature involving writing performance, but have been less extensively validated. In one case, few device details were provided other than the fabrication supplier [22], whereas, in two other cases, the devices are evidently similar to tablet prototype 1 [7,19,23]. One of these latter two tablets has a tilt adjustment for ergonomic use in the magnet bore [19], whereas the other is fixed to face upwards [23]—both at some cost in ecological validity.

An alternative fMRI-compatible touchscreen technology has been introduced that records stylus trajectory using a light-emitting stylus and a light sensor [24]. Utilizing similar technology with a higher spatial and temporal resolution, the tablet system developed by Reitz et al. (Figure 4) has been developed for fMRI studies of dyslexia and dysgraphia, two learning disorders [25]. The stylus has a plastic plate on the tip for a perpendicular orientation between the stylus and tablet, while the tablet, on the other hand, is a non-metallic paper holding frame with color gradients that allow distinct light reflections for spatial resolution. Optical fibres are routed from the stylus tip to a light-tight box outside the magnet room, consisting of one fibre coupled with an LED to illuminate the pen position on the tablet, and another connected to the color sensor to receive the resulting reflected light. The stylus coordinates derived from the RGB value of the reflected light enable real-time digitization and display using a mirror mounted on the head coil for video feedback projection [25], similar to tablet prototype 1 [7]. As demonstrated by the preliminary results obtained from two 11 year-old participants, the Reitz tablet offers simplicity in design, inexpensive deployment, and sufficient spatial resolution to score writing and drawing tasks in patients with dysgraphia and dyslexia, albeit without VFHP [25].

## 3. Tablet Applications

Since the development of fMRI-compatible tablets, these devices have enabled multiple advances in the fields of neuropsychology, neurosurgery, and neurolinguistics. Research highlights are presented below.

### 3.1. Neuropsychological Tests

Neuropsychological tests (NPTs) are widely used in behavioural neuroscience and in the clinic to assess cognition and to assist in the differential diagnosis of brain dysfunction. Many are administered as pen-and-paper tasks, although newer NPTs are increasingly computerized. Despite the wide adoption of NPTs, the brain activity associated with NPT performance remains quite poorly understood, and has mostly come from studying patients with focal brain lesions [26,27,28]. Such studies are limited by the variability in spatial organization of brain anatomy, and the size and location of the lesion in individual patients. In addition to other trade-offs [29], these studies also make strong assumptions about the specific importance of the lesion areas, de-emphasizing the role of other networked brain regions that may be involved. With the tablet technology and fMRI, the relationship between brain activity and NPT performance can be examined in unprecedented detail: (a) informing and improving the current usage of NPTs by scientists and clinicians; (b) benefiting patients by improved detection and characterization of cognitive impairment; (c) assessing whether detection sensitivity and specificity can be improved by including fMRI and NPT data together, or by including enhanced test score metrics based on digitized tablet data; and (d) providing insight to design new and enhanced NPT methods in the future. With these long-term goals in mind, the existing tablet and fMRI work involving NPTs is summarized below.

#### 3.1.1. The Trail Making Test

The Trail Making Test (TMT) is a pen-and-paper test that is used very widely in clinical and research settings to assess cognitive processes such as visual search, visual planning, visuomotor control, as well as attention and memory [26,28,30]. The TMT consists of two parts (A and B), each of which involve linking a total of 25 randomly placed items in ascending order: part A involves linking numbers (1-2-3-4-5...); part B, which is more challenging, involves linking numbers alternating with letters (1-A-2-B-3-C…). An example of tablet prototype 2 TMT performance is shown in Figure 5. Each part is typically scored by measuring the completion time and recording the number of errors. Other metrics, such as the ratio of completion times (B/A) are sometimes used to de-emphasize the visuo-motor aspects of performance and to emphasize the cognitive aspects [26,31,32].

In the development of tablet prototype 1, the TMT was used as a proof-of-concept fMRI experiment to demonstrate tablet capabilities and applicability to NPT research [7]. Tablet-based fMRI experiments were conducted involving two participants, showing that the measured mean behavioural response times for both TMT parts were largely improved compared to those obtained previously with a fibre optic-based drawing device [14], with values approaching the median normative scores for the pen and paper TMT [33,34]. Consistent with the previous fMRI results, the TMT-B vs. TMT-A contrast revealed increased left hemisphere activations in the middle frontal gyrus, superior frontal gyrus, middle temporal gyrus, and the superior parietal lobule, which are associated with executive function, spatial attention, and visuomotor control [7,14,35]. In addition, tablet prototype 1 was then used in methodological research to implement a block-design version of the TMT task, illustrating the improvements on fMRI detection sensitivity that can be obtained by optimizing “pre-processing pipeline” choices (the collection of image processing algorithms typically used to de-noise fMRI data prior to estimating brain activity) on a participant-by-participant basis, rather than applying a fixed set of choices across participants [36,37]. Age-related declines in TMT brain/behaviour relationships were also revealed using this methodology [36].

Subsequently, the neural correlates of the TMT were studied in young healthy adults to investigate the effect of tablet prototype on brain and behavioural measures [15]. Fixed-duration trials of the TMT were conducted during fMRI of two separate groups of eleven participants, with one group using tablet prototype 1 configuration (no VFHP) and the other using prototype 2 (VFHP). Both groups also performed the pen-and-paper TMT so that ecological validity could be assessed. Rather than evaluating completion times, as done previously [7,14,33,34], the study employed a new metric called seconds per link (SPL) to account for situations where participants did not complete a TMT trial within the fixed time limit [15]. The SPL values obtained with the tablet TMT were shown to be slightly increased (i.e., performance was slower) compared to the values obtained for the standard pen-and-paper TMT [15]. The ecological validity was judged to be good despite inevitable, minor differences between the testing environments inside and outside of the MRI system. Furthermore, tablet SPL values were significantly correlated with standard SPL values in TMT-A independent of tablet prototype, but not in TMT-B. Regarding brain activity, irrespective of tablet prototype, the most common mean activations in both TMT-A vs. control (visual fixation) and TMT-B vs. control contrasts were found in regions associated with somatosensory and motor processes, visual perception, imagined movement and visual search [38,39], as expected. Activation was also more left-lateralized for the TMT-B vs. TMT-A contrast with VFHP (prototype 2), compared to the case without VFHP (prototype 1), as expected. Areas engaged in executive function, motor planning, visual search, and performance monitoring were found to be active with VFHP [10,12,14,28,39,40]. The study concluded that both tablet prototypes were acceptable for assessing the neural correlates of TMT performance in young healthy adults. Tablet prototype 2 was found to be slightly more favourable, although the TabletCam introduced somewhat more obstruction of visual stimuli by the user’s hand than typically occurs during pen-and-paper administration [15].

Tablet prototype 2 has also been used in an fMRI study of TMT performance in a group of 36 healthy elderly adults (52 to 85 years old) [41] with administration of the pen-and-paper TMT for comparison. The tablet produced somewhat slower performance, consistent with the previous study, although significant positive correlations between tablet and pen-and-paper SPL values were found for both TMT parts, indicating good ecological validity. Poorer performance was observed with increasing age for both TMT parts, as expected. The fMRI activation maps showed activation patterns in both contrasts (TMT-A vs. control and TMT-B vs. control) that were generally consistent with previous TMT studies [7,15,26]. When exploring how these patterns of brain activity covaried with age (Figure 6), the older adults demonstrated less TMT-related brain activity compared to younger adults [41]. Importantly, this work provides initial normative data for future studies of TMT-related brain activity in elderly patient populations along the cognitive impairment spectrum (i.e., from mild cognitive impairment to more severe cases like Alzheimer’s Disease).

The initial tablet TMT fMRI work conducted to date provides a new picture of how multiple areas of the brain are engaged in TMT performance. Previously, it was thought that the TMT primarily assesses frontal lobe function and especially left-lateralized frontal regions when comparing performance of TMT-B to TMT-A [26]. Tablet fMRI studies reveal that many more brain regions are engaged when each TMT part is performed, implying that impaired TMT performance may be associated with damage to multiple regions in the brain, and their interconnections [7,15]. The most recent fMRI TMT study involving elderly participants has also elucidated functional roles of specific brain regions engaged in TMT performance, beyond the first step of simply determining the brain regions engaged and their locations [41]. Pursuing such research is important from a basic science perspective, and to help inform clinicians about how to interpret TMT performance.

#### 3.1.2. The Clock Drawing Test

The clock drawing test (CDT) is another widely used NPT, probing visuospatial processing, semantic memory, planning, and executive function. The CDT is primarily used to screen for dementias and other neurocognitive disorders such as Huntington’s disease [42,43]. The test involves instructing participants to draw a clock with a specified time in three steps: draw a circle, fill in the numbers, and draw clock hands. The test is typically evaluated based on a standardized score and the completion time. Many scoring systems exist, but one that is commonly used in research settings assesses the clock face (R1), the numbers within the clock (R2), and the clock hands (R3) [43].

Brain activity and CDT performance were recently studied in the same cohort of healthy elderly adults as discussed immediately above and shown in Figure 6, using fMRI and tablet prototype 2, including performance of the standard pen-and-paper CDT for comparison [44]. For both the standard and tablet CDT, significant negative correlations were found between CDT total score (sum of R1, R2, and R3) and age (i.e., older participants performed less well than younger participants). A significant positive correlation was also found between the tablet CDT scores and the paper scores [44], suggesting that the tablet had good ecological validity when applied in this context. In the fMRI analysis, the contrast of CDT vs. control (visual fixation) revealed increased activity in regions associated with visual processing, visuospatial perception, executive function, working memory, motor performance and somatosensory processing [45,46,47,48,49]. The study also reported reduced CDT-related brain activity in older adults compared to younger adults, suggestive of neural changes underlying age-related reductions in CDT performance [44].

#### 3.1.3. Letter Cancellation Test

The letter cancellation test (LCT) is commonly used in clinics and in research to assess attention, visuomotor control, and response shifting [28,50]. The test involves crossing out (cancelling) a specified target letter in an array that is mostly filled with foil letters. Performance is assessed using completion time and the number of errors (commission and omission).

The prototype 2 tablet system was also used to study the neural correlates of the LCT in the same group of healthy middle-aged-to-elderly participants that performed the TMT and CDT, as described above [51]. The standard pen-and-paper version of the LCT was also performed for comparison. Performance was quantified using “seconds per hit” (SPH) values, indicating the average time each participant took to make a correct cancellation. For both standard and tablet administrations of the LCT, significant positive linear regressions between age and SPH were observed, indicating that older adults exhibited poorer LCT performance compared to younger adults. Correlations between the two administrations, however, were non-significant and SPHs were significantly elevated in the tablet LCT performance compared to the paper LCT performance. This lack of concordance indicates that, unlike the TMT and CDT tablet tasks, the tablet version of the LCT may be a less naturalistic approximation of the standard paper and pencil administration. This was likely because of the dense spacing of the letter array in comparison to the visual stimuli adopted in the other two tests, which led to a more pronounced visual obstruction effect when participants viewed their hand in the augmented reality environment of tablet prototype 2. Regarding the neural correlates, the contrast of LCT vs. control (visual fixation) revealed increased activity in areas associated with visuospatial attention, visuomotor control, visual search, and target detection [35,52,53]. The study also showed that older adults exhibited less LCT-related brain activity compared to younger adults, suggestive of neural changes underlying age-related reductions in performance [51].

Considering the collective fMRI findings for the TMT, CDT and LCT reported above, one striking feature is that the fMRI activations maps for each NPT show significant overlap in the various brain regions that were engaged. This may be partly due to the very similar mode of response required in each of the tests, helping to explain why these tests are quite sensitive, but somewhat lacking in specificity [28]. The sensitivity likely arises because damage to one or more of many brain regions, or their connections, could lead to performance decrements; the specificity problem arises because the tests probe many of the same brain regions, and many of these regions can be damaged by more than one brain disease.

To conclude this section, the existing tablet-based fMRI studies of NPTs are still quite preliminary. More research is needed to examine other specific NPTs, to determine which activated brain regions are critical to certain aspects of NPT performance, and which are simply “along for the ride” as a general result of stylus/tablet interactions. One enabling approach involves coupling tablet-based fMRI experiments with additional surrogate behavioural measures (e.g., using eye tracking to identify brain regions that play a role in visual processing) [54]. Another option involves undertaking tablet-based fMRI with targeted neural stimulation (e.g., transcranial magnetic stimulation) to investigate how NPT performance and brain activity are affected by excitation or inhibition of certain brain regions [55].

### 3.2. Neurosurgery

#### 3.2.1. Awake Craniotomy Application to Brain Tumours

The fMRI-compatible tablet technology has also been useful for preoperative planning and intraoperative brain mapping in “functional neurosurgery” research [56,57,58]. During awake craniotomy procedures for patients with brain tumours near language or motor processing regions, direct cortical electrical stimulation (DCES) is used to guide and maximize the extent of tumour resection, while minimizing the resection of surrounding normal tissues—and behavioural side-effects. In particular, administering simplistic language tasks such as overt number counting and object naming may not be sufficient to engage the full extent of brain areas involved in language processing and more sophisticated tasks may be useful in some cases. Furthermore, pre-operative fMRI of these patients can also be performed with the resultant activation maps used to identify DCES targets, and to confirm DCES findings. The tablet system was therefore adapted to enable pre-operative and intra-operative usage in writing tasks. The writing modality was chosen as it can be performed during both pre-operative fMRI and intra-operatively (thus producing the same brain activity), and because written responses are a natural form of human communication that activate language processing regions while avoiding the technical difficulties involved in robustly performing fMRI of overt speech [59,60].

Interestingly, awake craniotomies introduce constraints on writing and drawing tasks that are similar to those during fMRI. The head of the patient is pinned within a stereotactic head frame, and surgical draping usually makes it impossible for the patient to see their hands during the performance of motor tasks. Thus, based on the two previous prototypes, a version of the tablet system was developed involving multiple video cameras, a computer, and the touch sensitive tablet together with a split-screen monitor so that the surgical team could monitor the patient’s hand interacting with the tablet, the face of the patient, the “craniotomy window” showing the surgical field, and the visual stimuli and writing/drawing responses (Figure 7) [57]. In addition, the system was supplemented with a microphone to record overt responses when needed. The system was programmed to support several tablet-facilitated language tasks, such as number counting and phonemic fluency (recalling as many words as possible that start with a certain consonant, within 60 s), enabling the patient to respond either by overt speech or writing [57,61,62,63]. In related experiments, both the overt speech and written versions of the phonemic fluency task were found to produce similar behavioural performance as measured by the average words per minute, for a sample of 12 young healthy adults [62] and 45 patients with Parkinson’s disease [64]. In addition, the brain activity associated with the written version of phonemic fluency was found to be consistent with previous results of verbal fluency tasks [62,65,66,67]. As illustrated through four clinical cases to characterize apraxia and detect speech arrest, robust mapping results between preoperative fMRI and the intraoperative testing platform were observed for language production tasks facilitated by the tablet [57]. Such findings suggested that the tablet platform improved the neurosurgeon’s ability to characterize and detect language deficits, providing a robust language mapping tool with advantages over previous standard approaches. The tablet platform also enabled expansion of the test repertoire and improved test flexibility during intraoperative mapping, as well as standardizing behavioural tests in both pre- and intra-operative mapping [57].

Furthermore, the tablet tasks developed for intra-operative DCES were found to have reasonable fMRI test–retest reliability in cohorts of healthy adults and patients with brain cancer (n = 12 for each group) [56]. This was important to assess, as pre-operative fMRI requires interpretation of brain activity from a single participant (i.e., the patient), whereas most fMRI discovery research reports the average brain activity from many participants with sufficient sample size to detect significant effects from the small, noisy fMRI signals. A study has also been conducted to evaluate the spatial concordance between fMRI and DCES, which is crucial for using fMRI results to guide the intraoperative mapping procedure [58]. Significantly higher concordance was seen in motor mapping compared to language mapping, and concordance was increased by standardizing (i.e., minimizing behavioural differences in) tasks across fMRI and DCES using the tablet, versus use of non-standardized tasks [58].

#### 3.2.2. MR-Guided Focused Ultrasound Treatment of Essential Tremor

Tablet prototype 1 has also been adapted for transcranial magnetic resonance guided focused ultrasound (MRgFUS) treatment of essential tremor (ET), a prevalent movement disorder characterized by constant kinetic or postural tremor of the upper extremities. Recently, MRgFUS has become attractive for treating ET of the arm (as a lateralized procedure to improve activities of daily living) by ablating millimetre-sized brain targets noninvasively [68].

The Fahn-Tolosa-Marin Tremor Rating Scale (FTMTRS) is a qualitative/semi-quantitative rating scale for assessing tremor severity after MRgFUS, including subjective reports of sensorimotor side effects and residual deficits [69,70,71,72]. The standard version of the test requires clinicians to be present in the magnet room during MRgFUS treatment to move the patient out of the MRI system, assess tremor, move the patient in, and clear the room—a time-consuming process when performed repetitively. Reducing the time for FTMTRS testing is important, as the MRgFUS procedure takes several hours for targeting and treatment using multiple sonications. The inter-rater reliability is also poor in standard FTMTRS [73,74,75]. Thus, an adaptation was developed using tablet prototype 1 to help overcome these limitations by improving test accuracy and efficiency, and by supplementing the current scoring with more quantitative and objective testing procedures [76].

Consisting of three drawing tasks—a large spiral, a smaller and tighter spiral, and three separate straight lines—the tablet FTMTRS was administered pre- and post-MRgFUS outside of the MRI system as proof-of-concept. (Further work will be required to make the MRgFUS equipment more compatible with visual stimulus presentation, and the preliminary results suggest that this may have merit.) Utilizing the digitized drawing signal recorded by the tablet, a spectral analysis of stylus speed was conducted to differentiate high-frequency tremor (4–16 Hz) from low-frequency voluntary movement [74,76,77], as shown in Figure 8. Compared to pre-operative drawings, the high frequency tremor signature in the speed spectra of 12 patients with ET was significantly and consistently reduced in post-operative drawings made by the treated dominant hand, but not the untreated non-dominant hand. This suggests that the tablet can be used to quantify ET and thus assess treatment efficacy in an objective manner, supplementing more qualitative assessments [76].

These initial studies suggest that tablet systems can provide a versatile behavioural testing platform to assist in functional neurosurgery applications. As the sample sizes in the above-mentioned studies are small, more tablet research is needed to develop adapted versions of clinical tests, to examine their efficacies, and to validate their usage by assessing test–retest reliability as well as behavioural and brain activity relationships, in larger groups of patients.

### 3.3. Neurolinguistics

Language processing has been traditionally studied by fMRI using visual or auditory stimuli for reading and listening, and button press responses to make language-related judgements [78,79]. However, the response mode is not ecologically valid and limits the aspects that can be probed. As described above, fMRI-compatible tablets can circumvent this problem because writing is a natural form of human communication that involves language processing in close association with visuo-motor control of hand movements. Various tablet studies have thus investigated how writing tasks are processed by the healthy brain or by the diseased brain [19,22,80,81].

One such study was performed using a tablet very similar to prototype 1 [19]. A total of eighteen young healthy adults were instructed to write letter pairs and digit pairs on the tablet according to presented auditory stimuli. Analyses contrasting the writing letter vs. control (holding the pen still) and writing digits vs. control revealed writing-related increases in activity consistent with previous writing research using DCES and fMRI [82], and with activation likelihood estimate meta-analysis [83,84]. The fMRI activation map of the writing letters vs. writing digits contrast revealed increased activity in regions linked to motor control [85,86], phoneme (smallest sound unit)-grapheme (smallest writing unit) conversion [87], and phonological processing of letters [88]. In another fMRI study, tablet-based written naming, oral spelling and drawing tasks were administered to sixteen healthy young adults [22]. The graphemic/motor frontal area (GMFA) was found to exhibit left lateralized activity during written word production (Figure 9, bottom row) and bilateral activity during drawing (Figure 9, top row). Writing-related positive brain activations were also observed [22].

Tablet methods have also been used to study the differences in writing-related brain activity across different languages and writing systems. For example, Mandarin Chinese has a logographic writing system, whereas Germanic languages like English and Romance languages like Italian have an alphabetic writing system. Recently, tablet prototype 1 was employed in several fMRI studies to examine the neural correlates of Chinese writing [89,90,91]. In an fMRI study of the brain activity associated with orthographic access during Chinese writing in 34 healthy young adults, a delayed copying task on high and low frequency Chinese characters was administered [89]. Correlations between tablet writing metrics indicated that items that require longer movement planning time also take longer to write. Such results support the writing theory that central processes such as language processing and motor planning are related to peripheral processes such as movement processing and execution [92,93,94], indicating excellent ecological validity. Both the high-frequency vs. control (drawing circles) and the low-frequency vs. control contrasts revealed activations associated with the central and peripheral processes in writing [83,84,89].

In a related fMRI study, researchers further investigated the neural correlates of Chinese writing in 33 healthy young adults [90]. Instead of relying on visual stimuli of words, as in the previous delayed copying task, participants were instructed to write Chinese characters using the tablet according to auditory stimuli (sounds of words), and to ignore auditory stimuli during the control task (drawing circles). The findings also supported the central and peripheral processes of writing and showed both similarities and differences in brain activity compared to that observed in other fMRI studies involving alphabetic languages. The same group of investigators also explored sex differences in both the behavioural performance and neural correlates of Chinese writing [91], involving 53 healthy young adults performing tasks as described above [89]. Although widespread brain regions associated with writing were observed, consistent with previous writing studies [22,83], the effect of sex on behaviour and brain activity was inconsistent and equivocal—either due to insufficient ecological validity of the tablet, or various other factors requiring further research study.

In addition to fMRI studies of logographic writing, a device similar to the Longcamp tablet [19] was used to examine the letter production system in the brain as a key aspect of alphabetic writing, for a group of 14 young healthy adults [95]. Brain activations associated with the motor and visual component of the letter production were subsequently revealed. The Reitz tablet [25] was also employed to understand the neural correlates of impaired alphabetic writing in learning disorders such as dysgraphia and dyslexia, in a group of 40 children with a mean age of 12 [80]. The children were grouped based on previously established behavioural markers for impaired writing (dysgraphia) or word reading and spelling (dyslexia) [96]. Via fMRI and diffusion tensor imaging experiments, researchers found that compared to healthy controls, children with learning disorders showed less white matter integrity, which were likely implicated in their impaired performance of the writing and word spelling tasks during group assignment [80]. Subsequent work [81] showed that certain children responded well to writing instructions but nevertheless exhibited reduced white matter integrity and abnormal gray matter functional connectivity. The researchers, therefore, proposed that, for differential diagnosis of learning disorders, behavioural assessment should be paired with additional evidence-based assessment [97]. These findings have implications in neuroscience research, pediatric clinics, and education [81].

Lastly, the SMART TAB tablet was developed for task-based fMRI research in the field of writing rehabilitation [23]. In two separate experiments involving 44 healthy young adults, strong test–retest reliability for a sentence writing task was established, and moderately significant correlations between behavioural performance outside and inside the magnet were found, establishing acceptable ecological validity. The fMRI activation maps of writing vs. rest contrast in one representative participant revealed increased writing-related brain activity consistent with previous writing literature [23,83,84], showing promising initial results—although much more work remains to be conducted in this area.

In summary, specialized tablet devices are starting to enable quantitative and objective assessment of writing kinematics, enhancing neurolinguistic fMRI research. This approach is leading to improved understanding of the mental processes that support writing performance and promises to assist in the differential diagnosis of learning disorders and in studying the effect of rehabilitation outcome. The existing fMRI-compatible technology is now quite mature, although, in some cases, increased use of advanced tablets (such as the prototype 2 configuration) may be required to enhance ecological validity.

## 4. Conclusions

Functional MRI-compatible tablet devices have various research and clinical applications, including but not limited to neuropsychological testing, functional neurosurgery, and neurolinguistics. The tablets serve as an excellent platform to approximate many traditional paper-based writing and drawing tests for fMRI, neurosurgery applications such as DCES and MRgFUS, and to study writing performance and its neural correlates in the healthy and diseased brain. Utilizing the digitizing feature of the tablets, enhanced behavioural recording and performance metrics become possible to examine behavioural kinematics efficiently with greater detail than is possible with traditional paper-based tasks. Such tablet technology may have commercialization potential, therefore, with applications in research or clinical markets that involve functional neuroimaging. For example, tablet prototypes 1 and 2 are the subject of 3 US patents [98,99,100]. In the future, technical improvements to tablets may be considered, such as to circumvent how the hand obstructs visual stimuli during VFHP, and further improvements to minimize input lag, increase sampling rate, and increase spatial resolution of the touchscreen according to need. Nonetheless, the existing fMRI-compatible tablet devices demonstrate powerful capabilities and strong potential to expand the behavioural repertoire of task-based fMRI for discovery neuroscience and clinical neuroscience research. Finally, such research may also benefit from the use of suitable tablet technology combined with other functional neuroimaging modalities, such as electroencephalography, for a more comprehensive understanding of task-related brain activity.

## Figures and Tables

**Figure 1 sensors-21-00401-f001:**
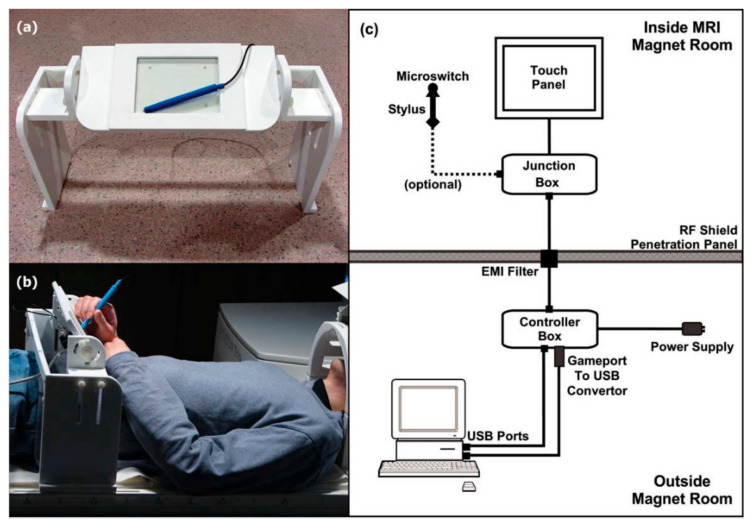
The fMRI-compatible tablet (prototype 1). (**a**) Close-up of stylus and tablet with mounting frame; (**b**) Tablet setup on the patient table, outside the magnet bore of an MRI system. A mirror was mounted on the head coil (just visible at the right) to enable real-time visual feedback of task stimuli and response via an fMRI-compatible projector (not shown); (**c**) Block drawing of the hardware setup inside and outside the MRI magnet room. fMRI = functional Magnetic Resonance Imaging; MRI = Magnetic Resonance Imaging; USB = Universal Serial Bus; EMI = Electromagnetic Interference. Taken from [7]. Copyright © 2010 Wiley-Liss, Inc., Hoboken, NJ, USA.

**Figure 2 sensors-21-00401-f002:**
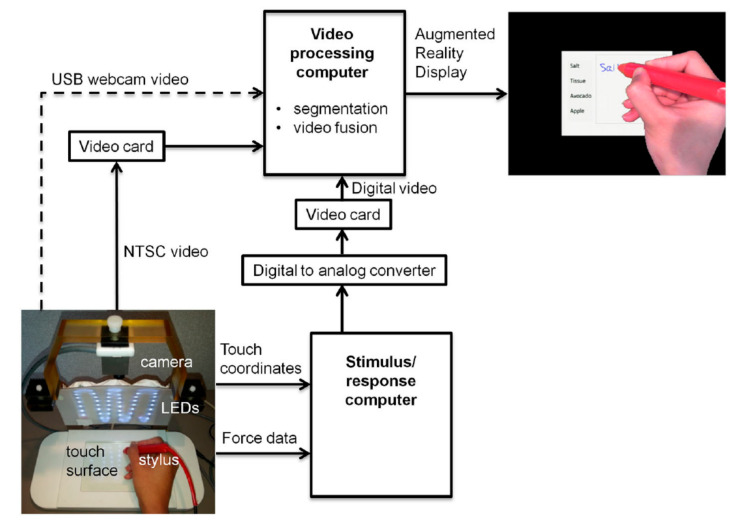
Block diagram for prototype 2 of the fMRI-compatible tablet system design. See text for details. USB = Universal Serial Bus; LED = Light Emitting Diode; NTSC = National Television System Committee. Taken from [17]. Copyright © 2015 Karimpoor et al.

**Figure 3 sensors-21-00401-f003:**
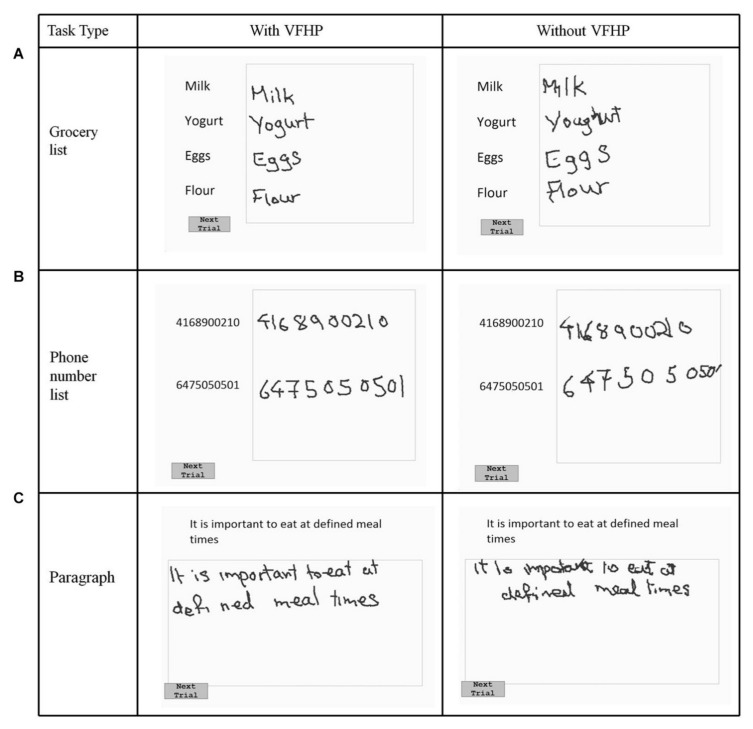
Visual stimuli and writing response sample of a writing experiment involving copying (**A**) a grocery list, (**B**) a phone number, (**C**) a paragraph using the fMRI-compatible tablet system with and without VFHP. The condition with VFHP achieved writing performance that was more clear, less cramped, and better positioned. VFHP = Visual Feedback of Hand Position. Taken from [17]. Copyright © 2015 Karimpoor et al.

**Figure 4 sensors-21-00401-f004:**
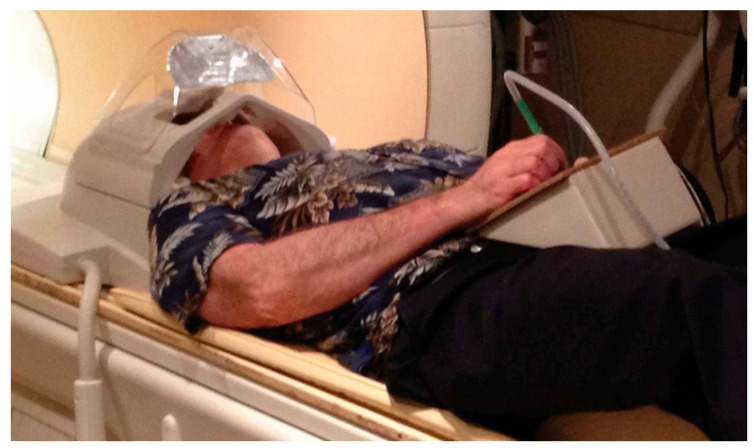
The fMRI-compatible Reitz tablet setup. The optical fibres are routed from the stylus tip to a light-tight box outside the magnet room. A mirror was mounted on the head coil to enable real-time visual feedback of task stimuli and response via video projection (not shown). Taken from [25]. Copyright © 2013 Reitz et al.

**Figure 5 sensors-21-00401-f005:**
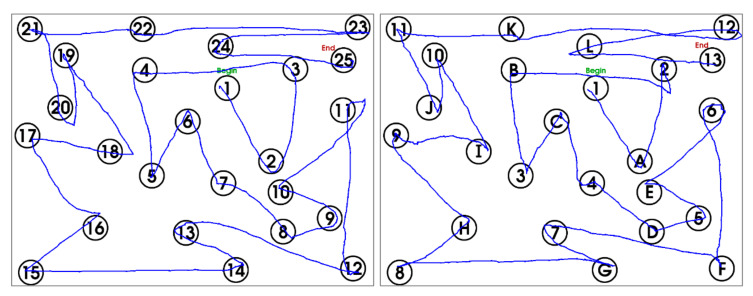
Example of tablet prototype 2 TMT behavioural performance. Left: TMT-A. Right: TMT-B. TMT = Trail Making Test.

**Figure 6 sensors-21-00401-f006:**
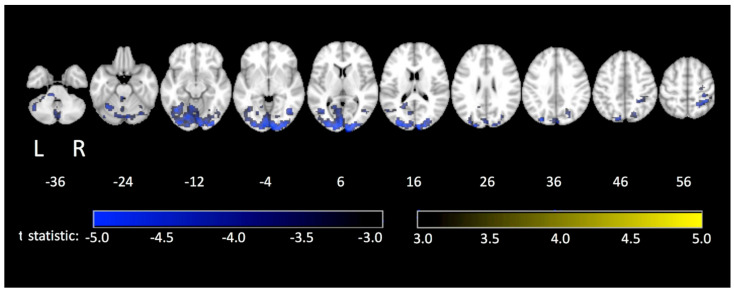
Regions with brain activity (TMT-A and TMT-B vs. control) that significantly covaries with age in a cohort of adults ranging from 52 to 85 years old. Numbers below the images represent the axial slice position in millimetres, in Montreal Neurological Institute (MNI) brain atlas coordinates. Taken from [41]. Copyright © 2020 Talwar et al.

**Figure 7 sensors-21-00401-f007:**
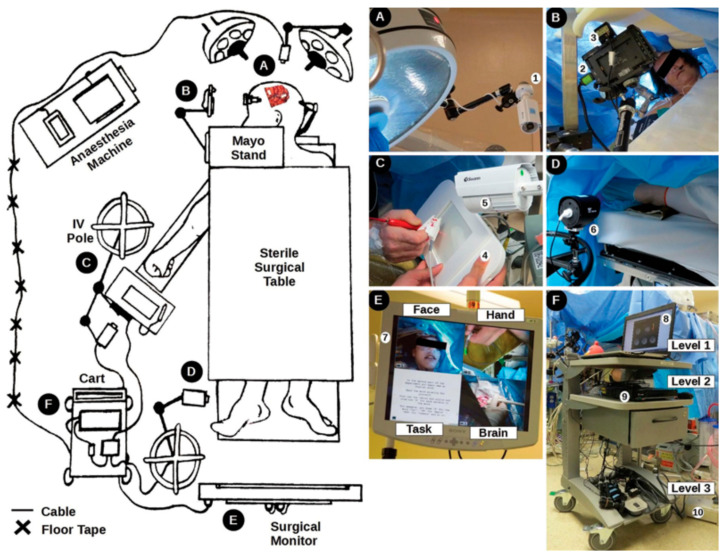
The intraoperative tablet testing platform setup in the operating room. (**A**) Surgical field; (**B**) Patient display; (**C**) tablet system; (**D**) foot camera; (**E**) video monitoring system; (**F**) intraoperative testing platform showing hardware in three levels for easy of setup and storage. IV = intravenous. Taken from [57]. Copyright © 2016 AANS. Permission to use this figure requires permission of JNS Publishing Group and is protected by US Copyright Law.

**Figure 8 sensors-21-00401-f008:**
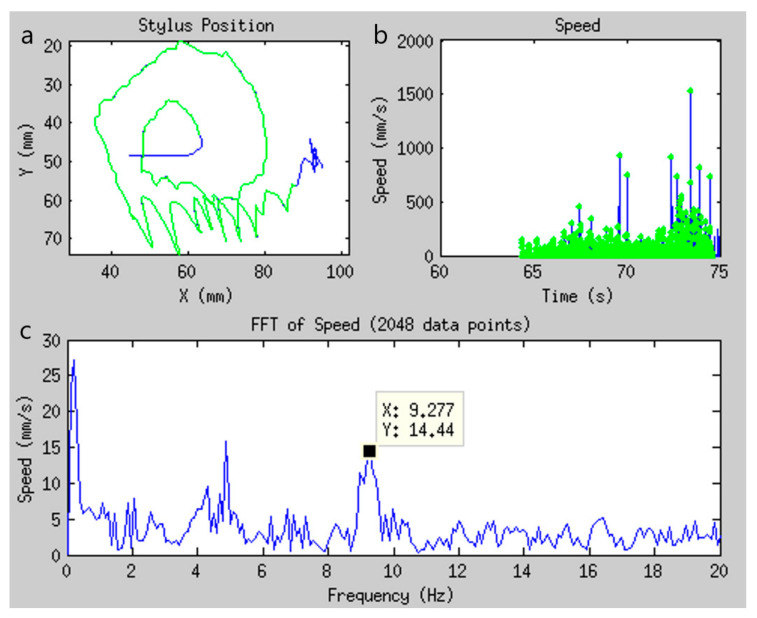
Tablet drawing signal analysis showing pre-treatment data for a patient with essential tremor. (**a**) Stylus position ink marks. Blue ink indicates raw cursor data, green indicates resampled cursor data which can be compared with the drawing screenshot (not shown); (**b**) Stylus speed over time (blue) with selected data points (green). Stylus speed was calculated by taking the derivative of stylus position; (**c**) The fast Fourier transform (FFT) of the selected speed data. Frequency peaks were selected for tremor evaluation. Taken from [76]. Copyright © 2017 Tam et al.

**Figure 9 sensors-21-00401-f009:**
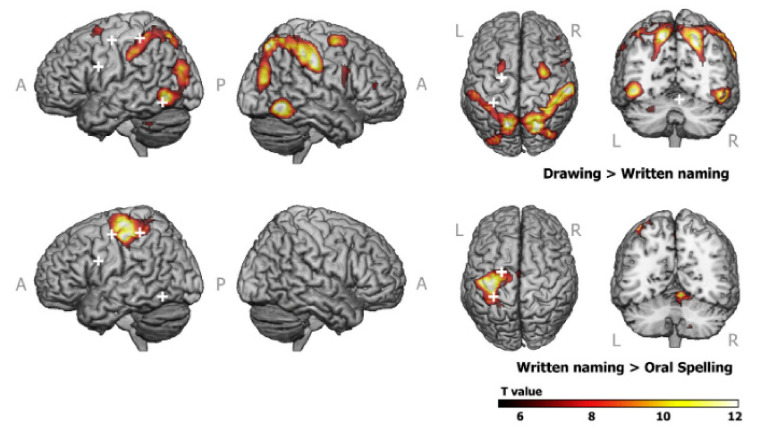
Activation maps of two major task contrasts in an fMRI study of writing behaviour. Top row: drawing vs. written naming; bottom row: written naming vs. oral spelling. A family wise error of *p* = 0.05 was used to threshold the activation maps. The white crosshairs indicate regions that were analyzed outside the scope of this review. See [22] for further details. A = anterior; P = posterior; L = left; R = right. Taken from [22]. Copyright © 2016 Elsevier Ltd., Amsterdam, NX, NLD.

**Table 1 sensors-21-00401-t001:** Pros and cons of techniques for studying writing and drawing behaviour during fMRI.

Technique	Tablet Prototype 1	Tablet Prototype 2	Mid-Air Finger Drawing [8]
Pros	Good ecological validity	Enhanced ecological validity (VFHP ^1^)	Naturalistic finger movement
	Digitized behavioural recording	Digitized behavioural recording	No hardware required
Cons	Hardware complexity	More hardware complexity	No interaction with writing surface
	More reliance on proprioception than Prototype 2	VFHP can obstruct visual stimuli	Participant receives no written performance feedback
			No behavioural recording

^1^ VFHP = visual feedback of hand position.

## Data Availability

Data sharing not applicable.

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
