# Peer review of "Tablet Technology for Writing and Drawing during Functional Magnetic Resonance Imaging: A Review"

_sensors, 2021, doi:10.3390/s21020401_

Round 1
Reviewer 1 Report
The review mainly describes different technological and applicative aspects related to the use of tablet-system for task-based fMRI with a special focus on the study of the brain activity associated with writing and drawing.
In particular, the paper includes 3 sections on 1) the optimization of fMRI-compatible devices and approaches; 2) the description of computerized "tablet system technology; 3) various fMRI applications where tablet systems have been utilized.
The review is well written and the reference literature was extensively considered.
I have just some minor comments to improve, hopefully, the paper.
Title: I would suggest including a specific focus on the writing/drawing tasks since all the review is devoted to that topic.
When referring to B0, please make sure it is actually a 0-zero and not an O. I think that in this version of the paper it is Bo and not B0.
Introduction: I would ask the authors to consider a cut of the part dedicated to the basic principles of MRI. I would just start directly with relevant information regarding fMRI.
Figures: Please make sure that the image quality is optimal.
Author Response
Thank you for the feedback! Please see the attachment.

Reviewer 2 Report
Authors reviewed various "Tablet Technology" for fMRI experiments, and also basics of MRI and fMRI. This study is adding new knowledge about "Tablet Technology" for fMRI experiments that is expanding the limits of conventional tasks for fMRI experiments. This study is helpful for neuroscientists, however, a few amendments should be made.
1.Although it is nice to summarize MRI basics and fMRI basics, this manuscript is supposed to be focused on tablet devices for fMRI experiments. Therefore, it would be better to make the first section (1. Introduction) shorter, perhaps you can leave out mechanisms of MRI as well as explanations for rs-fMRI because the tablet technique might not be usable for rs-fMRI experiments.
2.It is great if you could create a table of pros and cons of Prototype 1, 2, and a conventional technique for writing fMRI experiments to compare these three conditions. It can help us to understand how efficient the tablet devices.
3.Authors focused on neuropsychological tests (NPTs) in section 3 (3. Tablet Applications) and writing tasks in other sections. This is good, but the title is "Tablet Technology for Functional Magnetic Resonance Imaging: A Review." So, the title could be revised corresponding to their manuscript because they do not discuss or review general fMRI tasks.
4.They showed three examples of neuropsychological tests. Since these studies are important, but still preliminary, so could you summarize these three examples more? These parts are a little bit longer to get principle ideas or benefits of Tablet Technology for these studies.
5.I believe that explaining direct cortical electrical stimulation, ultrasound treatment, or other brain surgery methods is not an essential part of this manuscript. So, could you make those parts shorter? So that we can easily understand the usage of tablet technology for neurosurgery.
3.4. Neurolinguistics part could be more summarized. Generally, a detailed explanation for each example is not required for a review.
Minor
page 4 line 149
The authors mention that "usually over multiple collection periods each of at least 10 minutes to obtain reliable measurements of brain activity," however, there are many runs less than 10 min. So, this should be "about 10 min" rather than "at least 10 minutes."
There are several typos, such as follows.
page 14, line 537
"Functional MRI studies using tablet prototype 1 and 2 have have demonstrated that they are ..."
Author Response
Thank you for your feedback! Please see the attachment.

Round 2
Reviewer 2 Report
The authors have responded to all comments I suggested.